# The Influence of Oral Drotaverine Administration on Materno–Fetal Circulation during the Second and Third Trimester of Pregnancy

**DOI:** 10.3390/medicina58020235

**Published:** 2022-02-03

**Authors:** Paweł Rzymski, Katarzyna Maria Tomczyk, Maciej Wilczak

**Affiliations:** Department of Mother’s and Child’s Health, Gynecologic and Obstetrical University Hospital, Poznan University of Medical Sciences, Polna St 33, 60-535 Poznan, Poland; parzymsk@gpsk.ump.edu.pl (P.R.); mwil@gpsk.am.poznan.pl (M.W.)

**Keywords:** Doppler sonography, drotaverine, preterm birth, materno–fetal circulation

## Abstract

*Background and Objectives*: The study aimed to evaluate the effect of the oral administration of drotaverine on maternal and fetal circulation as measured by Doppler sonography in women with a risk of preterm birth. *Materials and Methods*: The present prospective study was conducted on 34 women with singleton pregnancy at 26–36 weeks of gestation. Doppler flow and pulsatility index (PI) assessments of the umbilical artery, fetal middle cerebral artery, and uterine arteries were performed before and 90–120 min after oral drotaverine administration. *Results*: There were no statistically significant differences between the Doppler assessment (PI Uma—umbilical artery, MCA—middle cerebral artery, and ltUta—left uterine artery) before drotaverine administration and 90–120 min after oral intake, but there were statistically significant differences between the PI assessment of the rtUta (right uterine artery, 0.55 vs. 0.75, *p* = 0.05) and the mean of the Uta (0.66 vs. 0.74, *p* = 0.03). For changes in the CUR (cerebro–umbilical ratio) and % changes in the CUR and mean PI of the Uta, there was no correlation with obstetric history, AFI (amniotic fluid index), gestation week, infertility history, systolic pressure, or diastolic pressure. There was a statistically positive correlation between changes in the CUR and % change in the CUR and body weight and in height. *Conclusions*: Drotaverine has no statistically significant influence on the MCA and Uma PI. The oral administration of drotaverine has an impact on PI rtUta and the mean PI Uta.

## 1. Introduction

The high risk of complications related to prematurity, including morbidity and preterm birth, is a serious obstetric problem worldwide [1]. Treatment is currently available in the form of a number of tocolytic agents, including beta-adrenergic agonists, calcium channel blockers, magnesium sulphate, prostaglandin synthetase inhibitors, and oxytocin receptor antagonists [2]. In Poland, since 30 October 2013, fenoterol has only been recommended for short inpatient tocolytic therapy in order to administer steroids that improve lung and respiratory system development [3]. Although tocolytic therapy is often recommended in those cases, there are additional commonly applied medications, such as drotaverine. Studies suggest that drotaverine therapy is safe during pregnancy; however, it remains an off-label treatment [4]. The drug is prescribed to smooth gastrointestinal and genitourinary muscle spasms [4]. Moreover, Drotaverine is a well-known antispasmodic. When it comes to obstetrics, the drug is considered to dilate the cervix, inhibit spasms, and shorten delivery time [5,6]. For instance, research indicates that it exerts a beneficial effect during labor [5,6]. However, besides labor, there are little data on the influence of administering drotaverine during pregnancy or its possible negative consequences on the fetus.

The role of drotaverine during labor has been well described in the literature [5,6]. The present study aimed to evaluate the effect of the drotaverine on the uterine arteries and fetal vessels as measured by Doppler sonography. To our knowledge this is the first study in the literature to measure the influence of drotaverine on materno–fetal circulation. All of participants were in their second or third trimesters of pregnancy and were at risk of preterm birth.

## 2. Materials and Methods

The present prospective study was conducted in the Department of Mother and Child Health in the Gynecological hospital in Poznań. A questionnaire and a consent were obtained from all the participants. Furthermore, all study protocols were approved by the Ethics Committee of Poznań University of Medical Sciences (decision No 1036/19, approval date: 07.11.2019.).

Thirty-four women of childbearing age at 26–36 weeks of gestation took part in the study. They had been hospitalized due to the risk of preterm birth during the period from October 2019 to June 2020. The inclusion criteria were regular uterine contractions, lower abdominal pain, changes in cervical dilatation, and a short cervix (less than 25mm), as determined during vaginal ultrasound examination. Patients who delivered within 7 days of admission to the hospital or who presented any symptoms of infectious disease were excluded from the study.

We collected data on obstetric history (at least one miscarriage), fertility (number of pregnancies), nicotinism, age, AFI (amniotic fluid index), systolic and diastolic pressure (RR systolic and diastolic), body weight, height, and the week of gestation.

Next, each patient took 80 mg of drotaverin (Sun-Pharm, Poland) orally. Then, Doppler flow and pulsatility index (PI) assessment of the umbilical artery (Uma), fetal middle cerebral artery (MCA), left and right uterine arteries (ltUta, rtUta) as well as fetal biometry were performed before drotaverine administration and 90 to 120 min after administration. A complete Doppler flow study in maternal and fetal circulation was assessed using the same Voluson 6 abdominal convex (GE Healthcare, Chicago, IL, USA).

As far as the umbilical artery is concerned, Doppler signals were obtained from a free loop of the umbilical cord and in repeated measurements from the same localization. Only regular waves were analyzed when the fetus was not breathing and moving to avoid artery flow impedance. The MCA Doppler assessment was proximally obtained, while the insonation angle was less than 30 degrees. The cerebral umbilical ratio (CUR = MCAPI / UmaPI) was calculated. A Doppler study of the uterine arteries was performed below the level of the apparent crossover with the external iliac artery. All of the sonography examinations in single patients were performed by one of two sonographers (20 and 6 years of Doppler sonography experience).

### Statistical Analysis

Standard Deviation (SD) and Mean were used to describe the collected data. Due to the normal distribution of the data, the paired T-test was applied to compare the changes in materno–fetal Doppler measurements. Alternatively, the Wilcoxon test was carried out to contrast the two groups when the data were not normally distributed. Pearson’s Correlation test was also used to assess the correlations between the two groups. Results were obtained using Sigma Plot 14.5 and Microsoft Excel. A *p*-value < 0.05 was considered statistically significant.

## 3. Results

Among all of the patients, there were no statistically significant differences between the pulsatility index (PI) assessment of the umbilical artery, fetal middle cerebral artery, and left uterine artery before drotaverin administration and 90–120 min after oral intake. There were statistically significant differences between the Doppler assessment of the right uterine artery and the mean of the left and right uterine arteries (Figure 1). The PI in those cases was higher after drotaverine administration, but the result still remained within the normal range.

We analyzed potential influencing factors. The changes in the Doppler cerebro-umbilical ratio (CUR) before and after drotaverine administration were calculated using the raw data and as the percentage of change. In terms of changes in the CUR and % change in CUR and mean pulsatility index of the uterine artery (PI Uta), there was no correlation with obstetric history, AFI, gestation week, fertility, systolic pressure (RR systolic), or diastolic pressure (RR diastolic). Additionally, there was a statistically positive correlation between changes in the CUR and % changes in the CUR and body weight and height (Table 1).

The same analysis was applied for the groups of smoking and non-smoking women. There were statistically significant correlations between changes in the CUR, % changes in the CUR, and height in the group of non-smoking women (Table 2).

After dividing the patients into two groups based on their week of gestation, the correlations between changes in the CUR and % changes in the CUR, body weight, and height were significant during the third trimester (Table 3).

Similar results were obtained while taking fertility into consideration (Table 4 and Table 5).

The correlations between changes in the CUR and % changes in the CUR and height were significant in the group of patients with a poor obstetric history (at least one miscarriage). These results were not observed in the group with an uncomplicated obstetric history (Table 4 and Table 5).

## 4. Discussion

Drotaverine hydrochloride is an antispasmodic drug that inhibits phosphodiesterase type IV, acting directly on smooth muscle cells, and it has a mild Ca-channel blocking effect [6,7]. Furthermore, it does not have an anticholinergic effect. It enables muscle relaxation and is thus commonly used to treat biliary renal, ureteric colic, and dysmenorrhoea as well as to shorten labor [4,8]. However, the administration of drotaverine during labor is not listed under the manufacturer’s indications for its use. Although the drug is commonly used in many countries, there is still not enough research on its safety for mother and child during labor [9,10,11,12]. The results of the meta-analysis revealed no serious adverse events in any of the studies after the administration of drotaverine during labor, although it should be noted that the authors recorded maternal and neonatal adverse events such as tachycardia, headache, nausea, or vomiting in only half the studies that have been conducted on this topic [9,10,11,12]. Serious complication, such as postpartum uterine atonia could also occur, but this has only been reported after intramuscular administration [13]. Therefore, further research should aim to provide more data to properly evaluate the drug.

In many countries, the drug is also administered to relieve the symptoms associated with pregnancy, such as abdominal pain or uterine contractions. In those cases, there is even less research evaluating the safety of drotaverine. According to the FDA, drotaverine is classified as being potentially risky, and hence, it should only be used when the benefits outweigh the risks of using it [9]. On the other hand, the modified classification indicates that drotaverine is possibly safe for use during pregnancy [9].

Many medical sources suggest that the bioavailability of the drug is similar after oral and intravenous administration [14]. Studies in rats mainly revealed absorption in the duodenum and intestine [15]. It was mainly metabolized in the liver and excreted by the non-renal route [15]. Still, little data are available on the influence of the drug on the human metabolism at pleasant. Nevertheless, in O Balii’s et al. research, it was pointed out that the high variation in the bioavailability of drotaverine HCl after oral administration may result in significant interindividual differences in terms of therapeutic response [14]. We chose the 90—120 min interval after the oral administration of drotaverine in order to perform our measurements at the point when the drug was at its highest potential concentration [16].

Drotaverine relaxes the smooth muscles [6,7]. As blood vessels also consist of smooth muscles, it would be interesting to examine whether the drug could dilate smooth muscles in blood vessels. Blood pressure reduction is the primary side effect of drotaverine. Although this side effect has been rarely reported, Sidrah Andleeb et al. revealed that the intravenous administration of the drug might cause hypotension in some patients [17,18]. Based on this, we hypothesized about the potential influence on materno–fetal circulation, but no important changes were noticed. Interestingly we observed an increase in the uterine vessels. Insufficient vasodilatation of the uterine arteries during pregnancy results in poorer utero-placental perfusion and potential fetal growth restriction. The activity of the large-conductance Ca2+ activated K+ channel (BK Ca) increased in uterine arteries during pregnancy in rats. Its inhibition reduces the uterine blood flow, which is an obvious adaptation to pregnancy. We could speculate that the mild Ca2+ blocking effect of drotaverine is similar to its adaptation in rats, where the inhibition of the Ca2+ channel increases the resistance index and pulsatility index [19].

Muhamad et al. noted a decrease in the spasm of the internal mammary artery and increases in blood flow after drotaverine was injected perivascularly during CABG (coronary artery bypass graft surgery) [20]. Moreover, vasodilation was also observed in the study of Zakharov et al. The researchers were examining the influence of intraarterial injections of No-spa on the dilation of muscle vessels that had been previously contracted by radiation. The results of their study indicate an increase in vessel dilation, even by as much as 30 % compared to the control group [21]. In our study, there was no increased blood flow through the left uterine artery, the umbilical artery, or the middle cerebral artery. However, we observed a statistically significant change in the pulsatility index of the right uterine artery and the mean pulsatility index of both uterine arteries after an 80 mg oral intake of drotaverine. Due to the higher pulsatility index after drug administration, the blood flow through the placenta might be diminished, but it probably has no clinical importance.

Maternal and fetal circulation is significant for fetal growth and proper pregnancy development. Uterine blood flow disorders cause pregnancy-induced hypertension, fetal growth retardation, and even stillbirths. Even though the impact of drotaverine on maternal and fetal circulation has not yet been examined, the drug is often used off-label to soothe uterus muscle spasms.

Our preliminary study revealed that the oral administration of 80 mg of drotaverine did not influence the Doppler flows (Uma, MCA). We noted a correlation between the time of drug administration and the pulsatility index of rtUta and the mean Uta. We also observed a clear correlation between the Doppler cerebro–umbilical ratio—CUR, and the height and bodyweight of a patient. The taller and heavier the patient, the greater the change in the CUR values before and after drotaverine administration. Additionally, the same correlation was found in the multiparous group. When it comes to patients in the second and third trimesters, the latter group showed a statistically significant correlation. Furthermore, the difference between the CUR value before and after drotaverine administration in the group with an uncomplicated obstetric history as well as in the group of non-smokers was related to the height of the patients. This effect is of unknown significance, if any. It is also important that in none of our patients demonstrated CUR changes to abnormal values, suggesting fetal cerebral blood flow centralization. It is clinically important and justified to analyze this aspect of a relatively old drug, even in off-label treatment during pregnancy and labor.

To the best of our knowledge, drotaverine has no clinically relevant influence on maternal and fetal circulation. Some correlations were found to be stronger in the group of obese and tall patients. Therefore, future studies are necessary to shed more light on the drug’s effect on maternal and fetal circulation.

## 5. Conclusions

The oral administration of 80 mg of drotaverine did not influence the Doppler flows (Uma, MCA).There is a correlation between the Doppler cerebro–umbilical ratio—CUR, and the height and bodyweight of a patient after the oral administration of 80 mg of drotaverine.After an oral dose of 80 mg of drotaverine the cerebro–umbilical ratio (CUR) did not change to abnormal values, and there were no signs of fetal cerebral blood flow centralization.

## Figures and Tables

**Figure 1 medicina-58-00235-f001:**
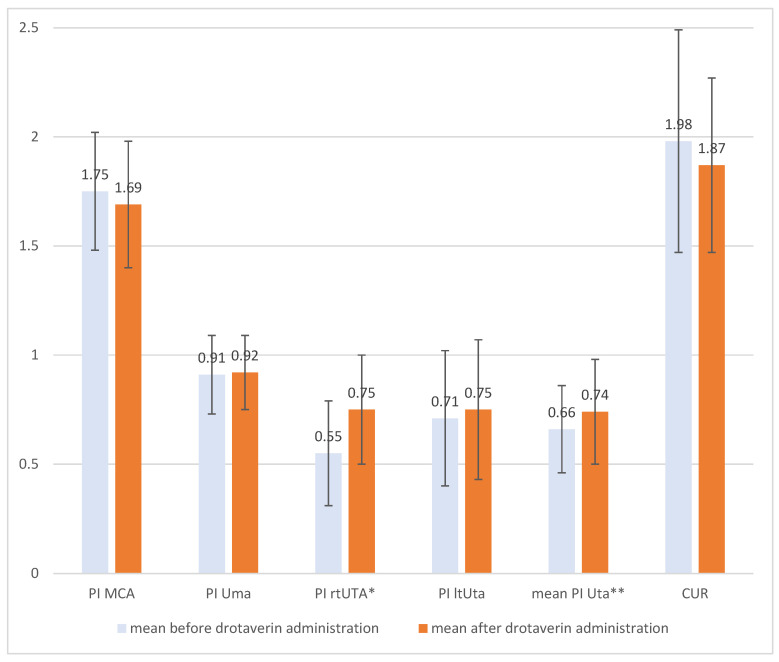
Comparison of pulsatility index (PI) before and after oral drotaverine administration (* *p* = 0.05, ** *p* = 0.03).

**Table 1 medicina-58-00235-t001:** The correlation between Doppler flows (concerning drotaverine administration) and selected patient features (statistically significant in bold).

		Time of Administration Drotaverine (90–120 min)	Obstetric History	AFI	Body Weight	Height	Gestation Week	Fertility	RR Systolic	RR Diastolic
Mean PI Uta	Correlation Coefficient	0.25	−0.06	0.23	−0.07	0.17	−0.27	0.13	−0.29	−0.24
	*p*-value	0.16	0.76	0.19	0.7	0.34	0.12	0.48	0.11	0.19
% change in CUR	Correlation Coefficient	0.21	−0.03	0.12	0.34	0.37	−0.07	0.09	−0.04	−0.08
	*p*-value	0.25	0.87	0.50	0.049	0.03	0.69	0.62	0.81	0.65
Change in CUR	Correlation Coefficient	0.12	−0.06	0.07	0.38	0.38	−0.13	−0.04	−0.05	−0.05
	*p*-value	0.51	0.76	0.72	0.03	0.03	0.47	0.83	0.77	0.79

**Table 2 medicina-58-00235-t002:** The correlation between Doppler flows (concerning drotaverine administration) and selected features in the non-smoking group of patients (statistically significant in bold).

		Time of Administration Drotaverine (90–120 min)	Obstetric History	AFI	Body Weight	Height	Gestation Week
Mean PI Uta	Correlation Coefficient	0.44	0.29	−0.34	−0.01	0.24	−0.19
	*p*-value	0.04	0.17	0.11	0.96	0.25	0.39
%deltCUR	Correlation Coefficient	0.16	−0.06	0.20	0.39	0.42	−0.18
	*p*-value	0.46	0.78	0.36	0.06	0.04	0.4
deltCUR	Correlation Coefficient	0.05	−0.07	0.13	0.40	0.45	−0.24
	*p*-value	0.82	0.8	0.54	0.06	0.03	0.26

**Table 3 medicina-58-00235-t003:** The correlation between Doppler flows (in relation to drotaverine administration) and selected features in the group of the patients during the third trimester (statistically significant in bold).

		Time of Administration Drotaverine (90–120 min)	Obstetric History	AFI	Body Weight	Height	Gestation Week	Fertility	RR Systolic	RR Diastolic
Mean PI Uta	Correlation Coefficient	0.22	0.21	0.33	−0.10	0.30	−0.26	0.14	−0.27	−0.29
	*p*-value	0.28	0.31	0.10	0.62	0.13	0.20	0.50	0.20	0.15
%deltCUR	Correlation Coefficient	0.30	−0.02	0.01	0.44	0.37	−0.11	0.12	−0.05	0.22
	*p*-value	0.13	0.94	0.97	0.03	0.06	0.58	0.57	0.83	0.29
0.29 deltCUR	Correlation Coefficient	0.18	−0.05	0.02	0.47	0.40	−0.16	−0.01	−0.05	−0.05
	*p*-value	0.37	0.83	0.91	0.02	0.04	0.43	0.95	0.81	0.8

**Table 4 medicina-58-00235-t004:** The correlation between Doppler flows (concerning drotaverine administration) and selected primiparous features (statistically significant in bold).

		Time of Administration Drotaverine (90–120 min)	Obstetric History	AFI	Body Weight	Height	Gestation Week	RR Systolic	RR Diastolic
Mean PI Uta	Correlation Coefficient	0.17	−0.38	0.42	−0.06	0.61	−0.26	−0.57	−0.47
	*p*-value	0.59	0.21	0.18	0.85	0.04	0.20	0.05	0.12
%deltCUR	Correlation Coefficient	−0.05	0.30	0.14	0.33	−0.44	−0.35	−0.12	−0.15
	*p*-value	0.88	0.35	0.66	0.30	0.15	0.26	0.72	0.63
deltCUR	Correlation Coefficient	0.01	0.24	0.18	0.37	−0.42	−0.34	−0.13	−0.20
	*p*-value	0.97	0.45	0.57	0.24	0.17	0.28	0.70	0.54

**Table 5 medicina-58-00235-t005:** The correlation between Doppler flows (in relation to drotaverine administration) and selected multiparous features (statistically significant in bold).

		Time of Administration Drotaverine (90–120 min)	Obstetric History	AFI	Body Weight	Height	Gestation Week	RR Systolic	RR Diastolic
Mean PI Uta	Correlation Coefficient	0.29	0.54	−0.44	−0.07	0.05	−0.36	−0.16	−0.12
	*p*-value	0.21	0.01	0.048	0.75	0.83	0.11	0.49	0.61
0.61%deltCUR	Correlation Coefficient	0.18	−0.14	0.19	0.48	0.45	−0.12	−0.05	−0.08
	*p*-value	0.43	0.56	0.41	0.03	0.04	0.62	0.84	0.74
0.74 deltCUR	Correlation Coefficient	0.11	−0.14	0.12	0.51	0.48	−0.15	−0.05	−0.02
	*p*-value	0.64	0.53	0.61	0.02	0.03	0.52	0.85	0.92

## Data Availability

Not Applicable.

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
