# Peer review of "The Influence of Oral Drotaverine Administration on Materno–Fetal Circulation during the Second and Third Trimester of Pregnancy"

_medicina, 2022, doi:10.3390/medicina58020235_

Round 1
Reviewer 1 Report
Rzymski et al prospectively assessed effects of drotaverine between 26-36wks’ gestation and the results were equivocal. Both Doppler-flow USG and pulsatility index were studied but the abstract states significant changes were seen only in the latter. There are fundamental problems with this report in style and content, however it might be salvageable if properly reshaped, as follows:
Major concerns
The paper does not correctly frame current experience with the subject of study. For example, at the final sentence paragraph 1 (introduction) is stated ‘there is no data on the influence of administering drotaverine during pregnancy …’ This comment is misleading and incorrect; it seeks to exaggerate the dearth of available information. The problem is in fact covered in more than one Cochrane Pregnancy and Childbirth Trials Register, plus several other published papers.
Presentation of key data is also extremely unclear and appears internally inconsistent. In the abstract, the Doppler component is described as ‘no change’ before vs after, while statistically significant differences were observed for PI. But in the Results section, these claims seem to be contradicted. Which part, if any, can be believed?
The 2 serious errors above would normally block publication of such a work. However, the investigation did entail collection of useful clinical data with an adequate sample, which might warrant publication under different terms of reference: Build on what you do find, not on confirming a null hypothesis. Consider highlighting the comparison between Doppler and PI characteristics after treatment, describing why only certain vessels were responsive and why this might matter, and propose a theory to explain observed differences.
With some exceptions, tables and charts make their best contributions when presenting meaningful differences. Here, one small data paragraph supports five cluttered tables, sometimes with red ink for unknown reasons. Perhaps the authors could instead give a functional diagram of the utero-fetal unit with vessels labelled, illustrating where drotaverine did and did not result in significant change?
Minor issues
All manner of discussion points like biophysics of drug pharmacokinetics, heart bypass graft surgery etc which, while of passing amusement, actually add little. Instead, consider the suggestion above on how to pick out the interesting findings and showcase that. The paper must maintain its focus sharply on detailing the vascular ramifications of drotaverine on pregnancy.
Drotaverine today is a rather controversial medication which is not even sold in many jurisdictions. With so many safer medications available to reduce preterm birth risk, why any interest is now given to this old topic is worth discussing.
It would be helpful for a native English language colleague to rework the document to attain improved fluency in expressing ideas.
Author Response
We appreciate the valuable reviews. As the research of old drug is focused on unknown influence on doppler indices have little literature representation, we worked with Your suggestions to improve the manuscript. The major part of drotaverine research in pregnancy was performed during labor, obviously without the doppler analyses. They were focused on safety and labor course. Please find attached detailed answers to Your reviews:
Rzymski et al prospectively assessed effects of drotaverine between 26-36wks' gestation and the results were equivocal. Both Doppler-flow USG and pulsatility index were studied but the abstract states significant changes were seen only in the latter. There are fundamental problems with this report in style and content, however it might be salvageable if properly reshaped, as follows:
Major concerns
The paper does not correctly frame current experience with the subject of study. For example, at the final sentence paragraph 1 (introduction) is stated 'there is no data on the influence of administering drotaverine during pregnancy ...' This comment is misleading and incorrect; it seeks to exaggerate the dearth of available information. The problem is in fact covered in more than one Cochrane Pregnancy and Childbirth Trials Register, plus several other published papers.
- We reedited this issue, there is literature of course, even from Cochrane Library cited in the manuscript but it is still concerning the effect during labor. But additionally to explain we added 4 more citations to the manuscript focusing on this problem. We agree, that “no data on the influence …’ is far exaggerated, even if the literature deals not exactly with the doppler imaging during labor
Presentation of key data is also extremely unclear and appears internally inconsistent. In the abstract, the Doppler component is described as 'no change' before vs after, while statistically significant differences were observed for PI. But in the Results section, these claims seem to be contradicted. Which part, if any, can be believed?
- The imprecise information in abstract has been corrected, additionally we added impoertan and exact values pre and post drotaverine administration. The abstract was probably too short in communication to emphasize what is written in full text.
The 2 serious errors above would normally block publication of such a work. However, the investigation did entail collection of useful clinical data with an adequate sample, which might warrant publication under different terms of reference: Build on what you do find, not on confirming a null hypothesis. Consider highlighting the comparison between Doppler and PI characteristics after treatment, describing why only certain vessels were responsive and why this might matter, and propose a theory to explain observed differences.
- We speculated about the CA2+ channels, blockers and blood flow based on rat model and added this suggestion to the discussion (lines 177-184)
With some exceptions, tables and charts make their best contributions when presenting meaningful differences. Here, one small data paragraph supports five cluttered tables, sometimes with red ink for unknown reasons. Perhaps the authors could instead give a functional diagram of the utero-fetal unit with vessels labelled, illustrating where drotaverine did and did not result in significant change?
- Red letters were mistakes during edition. We assumed that full data in tables are of superior quality compared to diagram, but if it is needed we could work hard on preparing communicative and full of data diagram. We are only afraid if it will not reduce the data set, possibly important for other researchers
Minor issues
All manner of discussion points like biophysics of drug pharmacokinetics, heart bypass graft surgery etc which, while of passing amusement, actually add little. Instead, consider the suggestion above on how to pick out the interesting findings and showcase that. The paper must maintain its focus sharply on detailing the vascular ramifications of drotaverine on pregnancy.
- Thank You, we added more literature but there is still lack in the research focusing on what we exactly did. That’s why we hope the context and other relation to the drotaverine, pregnancy and arterial disease is justified. But we emphasized closed aspects.
Drotaverine today is a rather controversial medication which is not even sold in many jurisdictions. With so many safer medications available to reduce preterm birth risk, why any interest is now given to this old topic is worth discussing.
- We added the paper discussing the Ca2+ blockers in rats’ pregnant uterine arteries blood flow based on doppler imaging. Some speculations concerning human similarities has been added.
It would be helpful for a native English language colleague to rework the document to attain improved fluency in expressing ideas.
- Yes, we will work with our colleague native speaker in English, but please let us some additional days, if the manuscript accepted.
Once again we would like to thank You for the time spent on the review. We prefer to discover something new rather than replay the well known research, that’s why we face some difficulties. The exact literature concerning utero-placental doppler imaging is missing, but we still hope this research adds something new to our clinical understanding of pregnancy.
Sincerely Yours
Katarzyna Tomczyk MD PhD
Assoc Prof Pawel Rzymski MD PhD

Reviewer 2 Report
The following points should be corrected to improve the quality of the work:
Abstract, insert the full of all acronyms present in the abstract.
SECTION 2. 3TH PARAGRAPH, use the full term of acronym 'AFI'.
Section 2. In the last paragraph, insert how it is calculated CUR.
Graphically adjustment of the tables (e.g width of the column) is required.
Section 4, Paragraph 1 citation 11, insert the type of adverse events (AE) of drotaverine, in the paper, you cited the following AE are reported: tachycardia, headache, nausea, and vomiting (no serious), but even atonic postpartum hemorrhage (serious). Specify that drotaverine was administered intramuscularly and not orally.
Section 4 paragraph 5, reduce the dimension of the word 'CABG'.
Author Response
We appreciate the valuable reviews. Please find attached detailed answers to Your reviews:
The following points should be corrected to improve the quality of the work:
Abstract, insert the full of all acronyms present in the abstract.
- done
SECTION 2. 3TH PARAGRAPH, use the full term of acronym 'AFI'.
- done
Section 2. In the last paragraph, insert how it is calculated CUR.
- done
Graphically adjustment of the tables (e.g width of the column) is required.
- we resigned to adjust them, we hoper for ideal adjustment during the editorial transofrmation
Section 4, Paragraph 1 citation 11, insert the type of adverse events (AE) of drotaverine, in the paper, you cited the following AE are reported: tachycardia, headache, nausea, and vomiting (no serious), but even atonic postpartum hemorrhage (serious). Specify that drotaverine was administered intramuscularly and not orally.
- We added them to the section
Section 4 paragraph 5, reduce the dimension of the word 'CABG'.
- We do not know this issue – we see the CABG word properly in our WORD version (dimensions, fonts etc)
Once again we would like to thank You for the time spent on the review. We prefer to discover something new rather than replay the well known research, that’s why we face some difficulties. The exact literature concerning utero-placental doppler imaging is missing, but we still hope this research adds something new to our clinical understanding of pregnancy.
Sincerely Yours
Katarzyna Tomczyk MD PhD
Assoc Prof Pawel Rzymski MD PhD

Round 2
Reviewer 1 Report
If the work was read/edited for English clarity, the objective was not achieved. It cannot be published until proper language skills are put into place.
In the abstract, the term 'fertility' is used where 'infertility history' seems more appropriate.
Author Response
Dear Reviewer,
1.If the work was read/edited for English clarity, the objective was not achieved. It cannot be published until proper language skills are put into place.
The paper work has been checked by doctor Robert Kippen- native speaker
2. In the abstract, the term 'fertility' is used where 'infertility history' seems more appropriate.
The term "infertility history" is corrected.
All of the changes in the paper work was marked by "".
In References we add articles: 11,12,13 and 22.
I hope the manuscript improved according to all reviewers suggestions.
Sincerely Yours,
Katarzyna Tomczyk MD, PhD
Assoc Prof Paweł Rzymski MD, PhD

This manuscript is a resubmission of an earlier submission. The following is a list of the peer review reports and author responses from that submission.